# Learning Enhanced Protein Representations from Molecular Dynamics Trajectories via Temporal GNNs

**Sajal Chaurasia,**\* **Pengkang Guo,**\* **Bruno Correia, Pierre Vandergheynst**
École Polytechnique Fédérale de Lausanne
Lausanne, Switzerland
`{sajal.chaurasia,pengkang.guo,bruno.correia,pierre.vandergheynst}@epfl.ch`

## Abstract

Proteins are inherently dynamic entities, yet traditional deep learning approaches predominantly rely on static structural representations. In this work, we employ Temporal Graph Neural Networks (Temporal GNNs) to encode the temporal evolution of Molecular Dynamics (MD) trajectories. Using the MISATO dataset, we show that predicting directly from dynamic data outperforms static baselines on atomic adaptability prediction and binding site detection. To address the computational cost of dynamic models, we propose a feature-based transfer learning strategy that distills dynamic knowledge learned through self-supervised pre-training into dense node features. Augmenting static models with these representations improves performance across tasks while avoiding the need to train costly temporal models for each downstream application. Our results suggest that dynamic knowledge captured by MD simulations can be effectively compressed and transferred to static models, bridging the gap between the richness of physics-based simulations and the efficiency required for practical applications.

## 1 Introduction

Proteins are fundamental molecular machines, central to nearly all biological processes including catalysis, signal transduction, and cellular regulation. Traditionally, the structure-function paradigm has driven research, operating on the premise that a protein's three-dimensional arrangement of atoms dictates its biological role. However, proteins are not rigid structures; they are inherently dynamic entities that undergo constant conformational fluctuations critical for their function. These dynamic behaviors range from atomic vibrations to large-scale domain motions, often revealing cryptic binding sites or transient states that static crystallographic snapshots fail to capture. Consequently, relying solely on static structures ignores the flexibility and range of structural states that play key roles in molecular recognition and drug binding.

The study of protein dynamics has been accelerated by the increasing availability of large-scale Molecular Dynamics (MD) datasets. While static structures have long been available through repositories like the Protein Data Bank (PDB), large-scale datasets of dynamic trajectories and thermodynamic ensembles have historically been scarce. To address this gap, Siebenmorgen et al. (2024) recently introduced the MISATO dataset, combining quantum mechanical properties with extensive MD simulations for approximately 20,000 protein-ligand complexes. By providing over 170 μs of accumulated MD traces in explicit water, MISATO offers a valuable resource for training AI models to learn physical parameters such as flexibility and induced-fit effects. Alongside MISATO, resources such as ATLAS (Vander Meersche et al., 2023), PLAS-5k (Bharathi et al., 2022), and other trajectory repositories are making protein dynamics increasingly accessible to data-driven methods.

Despite the growing volume of MD data, incorporating dynamic information into deep learning remains challenging due to the high dimensionality and temporal complexity of MD trajectories. Current machine learning approaches for proteins, particularly Graph Neural Networks (GNNs),

---

\*Equal contribution.

predominantly rely on static structural representations (Sarparast et al., 2024; Li et al., 2025; Yuan et al., 2024; Mahbub & Bayzid, 2022). Temporal GNNs offer a natural solution: unlike standard GNNs, they are designed to process time-series graph data, capturing the sequential evolution of protein conformations and predicting dynamic properties directly from raw simulations. In addition, feature-based transfer learning provides a strategy to reduce the cost of dynamic modeling. By pre-training a dynamic model on a self-supervised task—in our case, atomic adaptability prediction—we can distill dynamic knowledge into reusable node features that enhance static models on downstream tasks.

In this work, we employ Temporal GNNs to encode the evolution of MD trajectories. Our contributions include:

- **Direct Prediction from Dynamics:** We demonstrate that predicting directly from Molecular Dynamics data using temporal GNN architectures outperforms static structural models across two tasks: atomic adaptability prediction and binding site detection.

- **Feature-Based Transfer Learning:** We propose a transfer learning strategy that pre-trains a Temporal GNN on a self-supervised objective (atomic adaptability prediction), extracts learned node representations, and uses them to augment static models. This avoids training task-specific dynamic models while improving performance on downstream tasks.

## 2 RELATED WORK

**Temporal Graph Neural Networks.** Temporal GNNs provide a framework for processing sequential graph data, capturing both spatial topology and temporal evolution. Seminal work in this area, such as the Graph Convolutional Recurrent Network (Seo et al., 2016), combined spectral graph convolutions with LSTM units, while Temporal Graph Networks (Rossi et al., 2020) introduced memory modules to model continuous-time dynamic graphs. The ROLAND framework (You et al., 2022) proposed re-purposing static GNN architectures for dynamic settings by treating node embeddings at different layers as hierarchical states that update recurrently. In our work, we build on these architectures to model the temporal evolution of atomic coordinates in MD trajectories.

**Integrating Dynamics into Protein Models.** Recent work has sought to overcome the limitations of static protein representations by incorporating dynamic information. Guo et al. (2025) proposed a static-dynamic fusion approach that constructs a heterogeneous graph by augmenting static distance edges with dynamic correlation edges derived from MD simulations. Zhang & Vitalis (2025) introduced Nearl, an automated pipeline that extracts local and global motion features from trajectories and voxelizes them into 3D grids for processing by 3D-CNNs. In contrast, we propose two complementary strategies: applying Temporal GNNs to predict directly from sequential MD data, and a transfer learning approach that distills dynamic knowledge from a self-supervised model into reusable node representations for static models.

## 3 METHODS

### 3.1 GRAPH CONSTRUCTION

We represent a protein complex as a graph $G_t = (V, E_t)$, where $V$ is the set of nodes and $E_t$ is the set of edges at time frame $t$. The MD trajectory provides atomic coordinates for $T = 100$ frames. Prior to graph construction, all frames are aligned to the initial structure via rigid-body superposition to minimize the root-mean-square deviation (RMSD) between equivalent atomic positions, removing global translations and rotations. This alignment step has been shown to improve prediction accuracy by isolating internal conformational changes (Guo et al., 2025). For static modeling, we construct the graph from the first frame ($t = 1$) only, while for dynamic modeling, we use the full sequence $\{G_t\}_{t=1}^{100}$.

### 3.1.1 NODE DEFINITION AND FEATURES

Nodes are defined at the granularity required by each task:

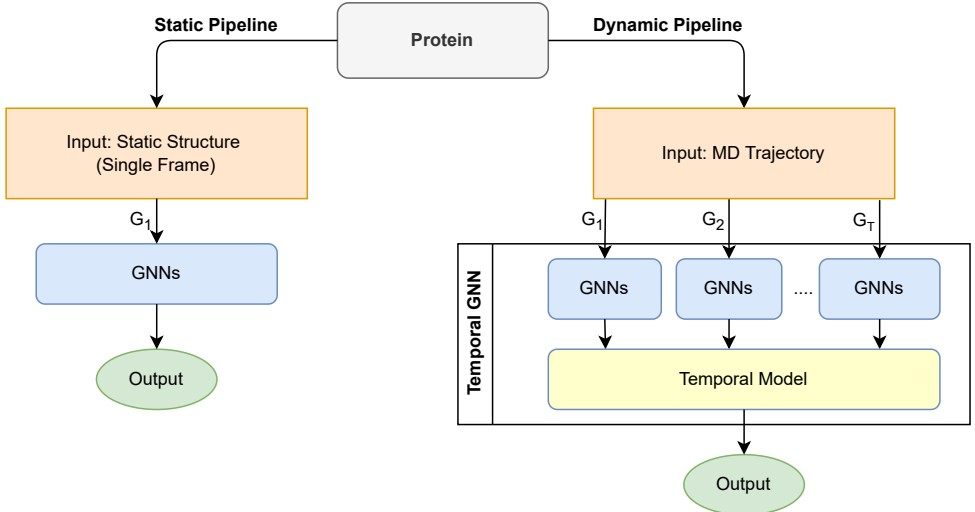

Figure 1: Overview of static and dynamic modeling pipelines. **Left:** The static pipeline processes a single-frame graph $G_1$ through a GNN. **Right:** The dynamic pipeline processes the full trajectory sequence $\{G_t\}_{t=1}^{T}$ through per-frame GNNs followed by a temporal model operating independently on each node's sequence.

**Atomic Adaptability Prediction.** Each node represents a non-hydrogen atom. Node features include: (1) a 10-dimensional one-hot encoding of atom type, (2) a 21-dimensional one-hot encoding of the parent residue type, (3) a normalized scalar indicating bond distance from the backbone, derived from PDB atom nomenclature, and (4) a normalized count of neighboring atoms within 4.5 Å, computed per frame to capture local density fluctuations.

**Binding Site Detection.** Each node represents a residue, positioned at its $C_\alpha$ atom coordinates. Node features consist of a 21-dimensional one-hot encoding of residue type.

### 3.1.2 EDGE CONSTRUCTION

We construct distance-based edges to capture spatial proximity at each frame:

$$E_t = \{(v_i, v_j) \mid d_t(v_i, v_j) < \tau_d\}$$

where $d_t(v_i, v_j)$ is the Euclidean distance between nodes at frame $t$ and $\tau_d$ is a distance threshold. We use $\tau_d = 4.5$ Å for atomic-level graphs and $\tau_d = 10$ Å for residue-level graphs, following established conventions (Bouysset & Fiorucci, 2021; Gligorijević et al., 2021).

### 3.2 NEURAL NETWORK ARCHITECTURES

As illustrated in Figure 1, we compare a static baseline with two dynamic architectures:

**Static Model.** We use the initial frame graph $G_1$ as input to a 5-layer GNN, where each layer is followed by normalization and ReLU activation. Final node embeddings are passed through a linear layer for node-level predictions.

**Dynamic Model (GCRN).** We adopt the GCRN architecture (Seo et al., 2016) to process the sequence $\{G_t\}_{t=1}^{100}$. A 5-layer spatial GNN stack first produces per-frame node embeddings, which are then fed into a temporal model operating independently on each node to capture temporal dependencies. A linear layer maps the final hidden states to predictions.

**Dynamic Model (ROLAND).** We adopt a ROLAND-inspired architecture (You et al., 2022) with five interleaved spatiotemporal blocks. Each block $\ell$ alternates between a spatial GNN layer and a

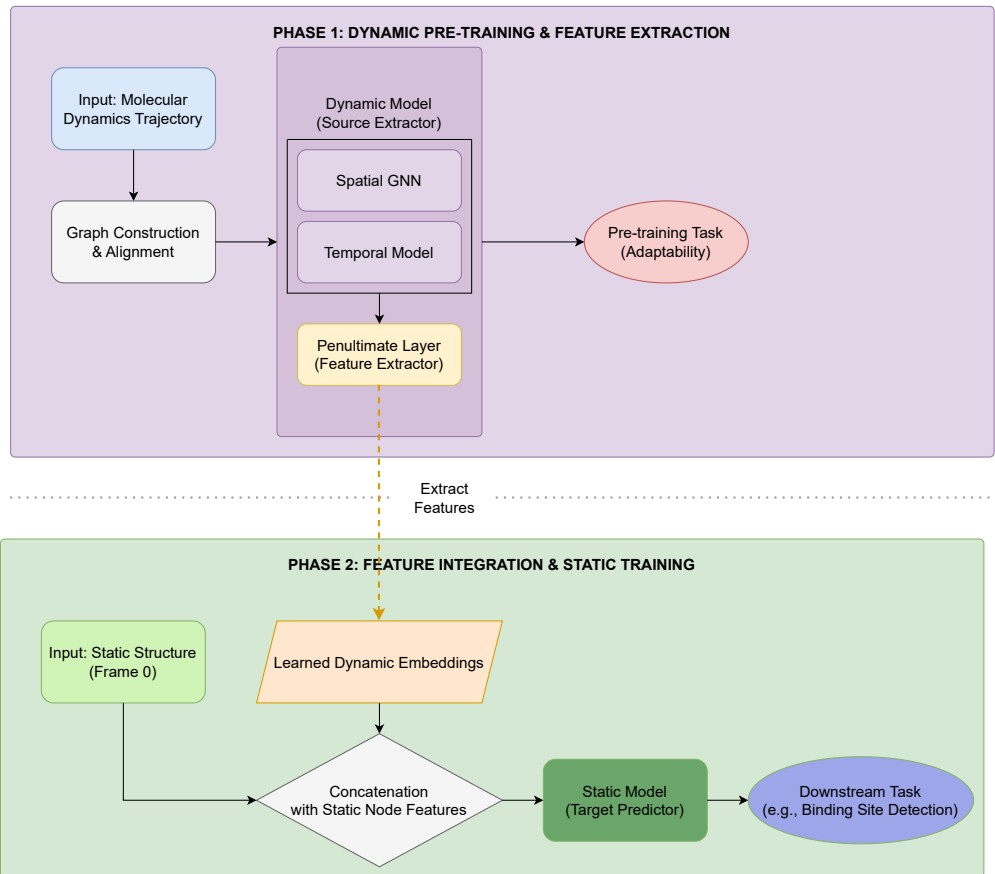

Figure 2: Overview of the proposed feature-based transfer learning strategy. **Phase 1:** A dynamic model (Spatial GNN + Temporal Model) is pre-trained on atomic adaptability prediction using MD trajectories, and node embeddings are extracted from the penultimate layer. **Phase 2:** The extracted dynamic embeddings are concatenated with static node features and used to train a static model on downstream tasks.

shared temporal model operating on each node's sequence:

$$H^{(\ell,t)} = \text{ReLU}(\text{Norm}(\text{GNN}_\ell(Z^{(\ell-1,t)})))$$

$$Z_i^{(\ell,t)} = \text{TemporalModel}(\{H_i^{(\ell,t)}\}_{t=1}^T)$$

where $Z_i^{(0,t)}$ denotes initial node features. Temporal model parameters are shared across all blocks, and the final state $Z_i^{(5,T)}$ is passed through a linear layer for prediction.

### 3.3 FEATURE-BASED TRANSFER LEARNING

To avoid training task-specific temporal models for each downstream application, we extract reusable node representations from a single pre-trained dynamic model (Figure 2). In the first phase, we pre-train a ROLAND model on atomic adaptability prediction as a self-supervised objective, then extract node embeddings from its penultimate layer. In the second phase, these embeddings are integrated into static models: for atomic adaptability transfer (T1-Static-Transfer), the embeddings are concatenated with the original atom-level features; for binding site transfer (T2-Static-Transfer), we select $C_\alpha$ atom embeddings and concatenate them with residue-type features. The augmented features are then used as input to a static GNN, requiring no temporal modeling for the downstream task.

## 4 Experiments

### 4.1 Dataset

We evaluate our approach on the MISATO dataset (Siebenmorgen et al., 2024), which contains 19,443 protein-ligand complexes derived from PDBbind (Su et al., 2018; Liu et al., 2017; Wang et al., 2005). Each complex undergoes semi-empirical quantum mechanical refinement and 10 ns molecular dynamics simulation using the Amber20 software package. The dataset is partitioned into training (80%), validation (10%), and test (10%) sets using BlastP-based sequence clustering to prevent data leakage from sequence homology.[1]

### 4.2 Setup

All model configurations use EGNN (Satorras et al., 2022) as the spatial GNN. For dynamic models, the temporal component is a GRU for atomic adaptability prediction and a Transformer (Vaswani et al., 2017) for binding site detection. Further implementation details and hyperparameters are provided in Appendix A.2.

### 4.3 Atomic Adaptability Prediction

Table 1: Atomic Adaptability Prediction Results. Node-level regression on the MISATO test set evaluated with RMSE, MAE, Pearson R, and Spearman R.

| Model | RMSE ($\downarrow$) | MAE ($\downarrow$) | Pearson R ($\uparrow$) | Spearman R ($\uparrow$) |
|---|---|---|---|---|
| T1-Static | 0.8894 | 0.5400 | 0.6743 | 0.6739 |
| T1-GCRN | 0.7560 | 0.4519 | 0.7700 | 0.7818 |
| T1-ROLAND | 0.7444 | 0.4605 | 0.7735 | 0.7781 |
| T1-Static-Transfer | **0.7286** | **0.4379** | **0.7805** | **0.7874** |

Atomic adaptability quantifies the conformational plasticity of each atom, defined as the mean squared displacement from its initial position across aligned trajectory frames (formal definition in Appendix A.1). We formulate this as a node-level regression task and report RMSE, MAE, Pearson R, and Spearman R on the test set.

Table 1 presents the results. The T1-Static baseline, which uses only the initial structural snapshot, achieves an RMSE of 0.8894 and a Pearson R of 0.6743. Both temporal architectures improve substantially: T1-GCRN and T1-ROLAND reduce RMSE by 15.0% and 16.3% respectively, with corresponding gains in correlation metrics. This confirms that processing the full trajectory sequence captures flexibility information that static snapshots miss.

The T1-Static-Transfer model, which augments static features with embeddings from the pre-trained dynamic model, achieves the best performance across all metrics (RMSE 0.7286, Pearson R 0.7805). This demonstrates that self-supervised pre-training on MD trajectories can distill dynamic knowledge into transferable features, improving static model performance without requiring temporal processing for each new downstream task.

### 4.4 Binding Site Detection

Binding site detection identifies residues that interact directly with ligands, a key task for understanding protein function and early-stage drug design. We formulate this as a residue-level binary classification problem, where a residue is labeled as a binding site if any of its atoms is within 10 Å of the ligand, following PDBbind conventions. Given the class imbalance (12.8% positives), we use weighted binary cross-entropy loss and report F1 score as the primary metric alongside precision, recall, and accuracy.

---

[1]We excluded graphs with more than 17,500 atoms ($\approx$0.10%) from training and validation due to GPU memory constraints.

Table 2: Binding Site Detection Results. Residue-level binary classification on the MISATO test set evaluated with F1 score, Precision, Recall, and Accuracy.

| Model | F1 Score ($\uparrow$) | Precision ($\uparrow$) | Recall ($\uparrow$) | Accuracy ($\uparrow$) |
|---|---|---|---|---|
| T2-Static | 0.3926 | 0.2954 | 0.5853 | 0.8204 |
| T2-GCRN | 0.4057 | 0.2987 | 0.6321 | 0.8164 |
| T2-ROLAND | **0.4642** | **0.3575** | **0.6620** | **0.8485** |
| T2-Static-Transfer | 0.4092 | 0.3035 | 0.6277 | 0.8203 |

Table 2 presents the results. The T2-ROLAND model achieves the best F1 score of 0.4642, an 18.2% improvement over the static baseline (0.3926), with gains across all metrics including precision (0.3575 vs. 0.2954) and recall (0.6620 vs. 0.5853). T2-GCRN also improves over the baseline (F1 0.4057), though the gap with ROLAND suggests that the interleaved spatiotemporal design is more effective than the sequential spatial-then-temporal approach for this task.

The T2-Static-Transfer model achieves an F1 of 0.4092, improving over the static baseline by leveraging dynamic embeddings from the pre-trained T1-ROLAND model. While the gain is more modest than the fully dynamic T2-ROLAND, it demonstrates that dynamic knowledge transfers across tasks: embeddings learned for atomic adaptability also carry information relevant to binding site detection.

## 5 CONCLUSION

In this work, we showed that dynamic models trained on MD trajectories significantly outperform static structural baselines on both atomic adaptability prediction and binding site detection, confirming that MD simulations contain rich temporal information that static snapshots alone cannot capture. To make this dynamic knowledge more accessible, we proposed a feature-based transfer learning strategy that pre-trains a dynamic model on a self-supervised objective and extracts reusable node embeddings to augment static models. This approach improved performance on both tasks— and on atomic adaptability prediction, it even surpassed the fully dynamic models—while avoiding the need to train task-specific dynamic models.

These results suggest that a single self-supervised dynamic model can both serve as a strong predictor and as a source of transferable representations for lightweight static models. Future work includes scaling to longer MD trajectories and exploring new self-supervised objectives and downstream tasks.

### ACKNOWLEDGMENTS

We thank the reviewers for their constructive feedback that helped improve the quality of this work. This research was conducted at the Laboratory of Signal Processing 2 and the Laboratory of Protein Design & Immunoengineering at École Polytechnique Fédérale de Lausanne.

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

## A  APPENDIX

### A.1  ATOMIC ADAPTABILITY DEFINITION

Atomic adaptability is defined as the mean-squared displacement of each atom from its initial position across aligned trajectory frames:

$$\gamma_x = \frac{1}{N_{\text{frames}}} \sum_{i=1}^{N_{\text{frames}}} \|r_{\text{ref},x} - r_{i,x}\|^2$$

where $\gamma_x$ is the adaptability of atom $x$, $r_{\text{ref},x}$ is its position in the reference frame, and $r_{i,x}$ is its position in frame $i$.

### A.2  IMPLEMENTATION DETAILS AND HYPERPARAMETERS

All models were implemented using the PyTorch Geometric library. We used the Adam optimizer ($\beta_1 = 0.9, \beta_2 = 0.999$) with a cosine annealing learning rate scheduler. The maximum number of training epochs was 100 for Task 1 and 200 for Task 2, with early stopping based on validation metrics. Due to the high memory footprint of all-atom temporal graphs (100 time steps), Task 1 required a batch size of 1, while the coarser residue-level graphs in Task 2 allowed a batch size of 4.

