# OpenReview forum: "Learning Enhanced Protein Representations from Molecular Dynamics Trajectories via Temporal GNNs"
_ICLR.cc/2026/Workshop/FM4Science — ICLR 2026 Workshop FM4Science Poster_

### Official Review · Reviewer_eMjo · 2026-02-18
**Efficient Integration of Protein Dynamics into Static Representations via Temporal GNNs**

**Rating:** 5
**Confidence:** 3

**Review:**

This paper addresses the limitation of static protein representations by employing Temporal Graph Neural Networks (Temporal GNNs) to learn from Molecular Dynamics (MD) trajectories in the MISATO dataset. The authors propose a feature-based transfer learning strategy that distills dynamic knowledge—captured through a pre-training task on atomic adaptability—into dense node features to augment static models.

**Pros**
- The motivation to incorporate dynamic conformational information into deep learning models is biologically well-founded, addressing the inherent limitations of static crystal structures in capturing protein function.
- The proposed feature-based transfer learning strategy effectively balances accuracy and efficiency by distilling expensive temporal dynamics into reusable static node features.
- Empirical results demonstrate that the augmented static model consistently outperforms the standard static baseline and achieves performance competitive with, or superior to, fully dynamic architectures.
- Utilizing the large-scale MISATO dataset allows for a robust evaluation of data-driven physics learning, bridging the gap between MD simulations and geometric deep learning.

**Cons**
- The architectural novelty is limited, as the core contribution relies on adapting existing Temporal GNN frameworks (GCRN and ROLAND) rather than designing specialized mechanisms for protein dynamics.
- Computational scalability remains a concern, evidenced by the necessity of a batch size of 1 for the atom-level task due to the high memory footprint of processing full trajectory graphs.
- The evaluation scope is relatively narrow, relying on only two downstream tasks (atomic adaptability and binding site detection) without assessing performance on other critical applications like affinity prediction or docking.
- The term "self-supervised" is used somewhat loosely, as the pre-training task is essentially a supervised regression on a geometric property derived from the trajectory rather than a standard pretext task like masking or contrastive learning.

---

### Official Review · Reviewer_dj1G · 2026-02-19
**Solid application of Temporal GNNs, good enough results**

**Rating:** 8
**Confidence:** 3

**Review:**

This paper uses Temporal GNNs (GCRN and ROLAND-style architectures) to learn from full MD trajectories, showing clear gains over static models on atomic adaptability and binding site detection. It further proposes feature-based transfer learning that distills dynamic knowledge into reusable node embeddings, improving static models without requiring temporal training per task.

The paper is clear and the feature-based transfer learning approach is simple, scalable, and effective (particularly strong for adaptability prediction where it even surpasses fully dynamic models), and the two tasks (regression + classification) and proper homology-aware splits evaluation is solid.

However, the static-transfer model improves over baseline but remains noticeably below the fully dynamic ROLAND model, raising questions about how much dynamic information is actually preserved.

---

### Meta-Review · Area_Chair_4Bcx · 2026-02-28

**Recommendation:** Accept (Poster)
**Confidence:** 3

**Metareview:**

This paper tackles the problem of protein representation learning. They argue that learning representations from Molecular Dynamics trajectories can outperform static baselines on atomic adaptability prediction and binding site detection. Reviewers agreed on the motivation and novelty of the work, and suggested enhancing the paper by extending the experiments to examine more critically how much dynamics information is actually learned, and to look at other downstream applications such as affinity prediction and docking.

---

### Decision · Program_Chairs · 2026-03-03

Accept (Poster)